# Comparison of Real and Simulated Fiber Orientations in Injection Molded Short Glass Fiber Reinforced Polyamide by X-ray Microtomography

**DOI:** 10.3390/polym14010029

**Published:** 2021-12-22

**Authors:** Rafał Żurawik, Julia Volke, Jan-Christoph Zarges, Hans-Peter Heim

**Affiliations:** Institute of Material Engineering, Polymer Engineering, University of Kassel, 34125 Kassel, Germany; zurawik@uni-kassel.de (R.Ż.); volke@uni-kassel.de (J.V.); heim@uni-kassel.de (H.-P.H.)

**Keywords:** Injection molding, injection molding flow simulation, model parameter study, X-ray microtomography, short glass fiber, polyamide 6, fiber orientation prediction, Moldflow^®^ Rotational Diffusion Model

## Abstract

During injection molding of short glass fiber reinforced composites, a complex structure is formed due to the fiber movement. The resulting fiber orientation can be predicted using various simulation models. However, the models are known to have inadequacies andthe influence of process and model parameters is not clearly and comprehensively described. In this study, the aforementioned model and process parameters are investigated to determine the dependencies of the individual influences on the real and simulated fiber orientation. For this purpose, specimens are injection molded at different process parameters. Representative regions of the specimens are measured using X-ray microtomography and dynamic image analysis to determine the geometric properties of the fibers as well as their orientations. Furthermore, simulations are performed with the simulation software Moldflow^®^ using different mesh types and densities as well as varying parameters of the MRD model to represent the real fiber orientations. The results show that different orientation areas arise in the samples, which cannot be represented with a simulation varying only one parameter. Several simulations must be carried out in order to represent flow regions occurring in the specimen as realistically as possible.

## 1. Introduction

Due to their lightweight potential combined with good mechanical properties, injection-molded short glass fiber reinforced thermoplastics (SGFRP) are being used in ever-increasing application areas. SGFRP are now being used in components exposed to high loads, which were originally reserved for metallic materials. The largest market is the automotive industry, where SGFRP are becoming increasingly popular to achieve the required weight and emissions reduction [1,2]. They are mainly used in covers, connector systems, bodywork, and structural components [3].

The low weight, cost savings, and ease of manufacturing geometrically demanding components in large quantities have led to a rapid increase in the use of SGFRP [4,5,6,7]. The use of glass-fiber-reinforced polyamide, e.g., for oil pans, bearings, and underhood applications, enables weight reductions of up to 50% compared to metal series components [3]. By reinforcing the polyamide with glass fibers, an increase in stiffness and strength is achieved with a simultaneous reduction in creep tendency, while at the same time, the density of the fiber-reinforced polyamide is only about 20% higher than that of the pure thermoplastic.

During the production of parts in the injection molding process, the fibers are oriented in specific directions due to shear and elongation stresses in the thermoplastic melt depending on the mold-related flow processes [8]. This fiber orientation has a direct influence on the mechanical properties and the anisotropic behavior of the composites since the composites exhibit significantly higher mechanical properties when loaded in the fiber direction due to the higher characteristic values of the fibers [9,10,11,12,13,14,15].

Many publications show that up to nine layers with different fiber orientations can be formed in injection-molded components, depending on their mold geometry [8,16,17]. These result from different flow directions during mold filling and process-related velocity and shear rate distribution over the component thickness. Generally, these layers can be subdivided into the three main layers of the skin, shear, and core layer, equivalent to the shear rate curve [18,19,20,21].

The random fiber orientation in the skin layer results from the solidification of the melt on the mold wall at low shear rates. In the shear zone (“shell layer”), the fibers are exposed to the highest shear due to the large shear rate gradient, which is why they align in the flow direction [18,22]. Towards the center of the part (“core layer”), the shear rate is reduced, the melt is only stretched by elongational stresses, and the fibers align primarily transverse to the flow direction [17,18,23,24]. The elongation of the melt in the core zone results from a diverging flow, which arises particularly in plates with high wall thicknesses in combination with gating systems with relatively small cross-sections [25]. Due to this flow, a biaxial elongation of the melt transverse to the flow direction and an equivalent fiber orientation is established [25,26].

In the case of thin-walled plates or components, depending on the process parameters, the shear layer can account for a relatively large proportion of 60–70% of the cross-sectional area, which is significantly larger than the area of the core layer with approx. 15% [27]. This suggests a large influence of the fibers present in these areas in the flow direction [18]. The example described here refers to the simple shear flow. However, for real geometries or for specimens taken at the edge, near the gate, or at the flow front, deviations from the typical above-mentioned fiber orientation distribution occur due to flow conditions during processing [23,28].

In order to comprehensively determine the correlation of injection molding process parameters and the process-dependent fiber orientation by experiment, a high level of testing, personnel, and material effort is required [29]. For a long time, fiber orientation in composites was determined by microscopic observation of the specimens, but X-ray Microtomography (µCT) is currently being used more frequently [12]. In contrast to micrographs, µCT allows a three-dimensional representation of the microstructure and thus also of the fibers in the composites without destroying them [30]. This also increases the accuracy of the determined geometrical and positional properties of the fibers or fillers [20,31].

For the evaluation of fiber orientation with µCT measurements, the tensor representation is often used for SGFRP. For this purpose, the tensor components a_11_, a_22,_ and a_33_ are plotted over the thickness of the specimen in a 3 × 3 matrix [20,21,23]. If the value for one of the components $a_{\mathrm{xx}}{,\;a}_{\mathrm{yy}}$, or $a_{\mathrm{zz}}$ is “1” during the evaluation, it means that the fibers are oriented in the direction of the corresponding axis [14,17] (see Figure 1). If the value of one of the main diagonal elements is “0”, the fiber is perpendicular to this direction, which results in a so-called two-dimensional random distribution [14]. With a determined value of a = 1/3, a three-dimensional random distribution is present [14]. Generally, fibers are never oriented strictly in one direction, resulting in a mix of all spatial directions [32].

To reduce the experimental effort, injection molding flow simulation (IMFS) software is used. The higher the prediction quality of the fiber orientation and length distribution of this software, the more accurate a prediction of the material structure and the resulting mechanical behavior can be made [33]. However, the currently used commercial IMFS software shows considerable differences to the real fiber orientation when calculating and displaying the process-induced fiber orientation [34]. For that reason, a comparison of the real and simulated fiber orientation is necessary, on the basis of which the simulation models and their parameters are to be adjusted, and the results are therefore approximated to the real structures.

The tensor representation of the fiber orientations is used in several software programs for IMFS as well as in all implemented models of Autodesk Moldflow^®^ used for simulations in this study. The models are based on Jeffrey’s model, which describes the motion of an ellipsoidal body in a Newtonian fluid and a simple shear flow with algorithms of translation and rotation [32,35]. The model assumes that the fibers in a simple shear flow aim for a stationary phase in which they are oriented in the flow direction according to the velocity field of the melt [36,37]. Folgar and Tucker extended Jeffrey’s model with the Isotropic Rotary Diffusion (IRD)-term, which includes the option to consider the interactions between the fibers in the melt using the fiber interaction coefficient C_i_ [8,38]. C_i_ is unique for each fiber-matrix configuration [27] and can be determined theoretically, numerically, or experimentally [39,40].

The Folgar–Tucker-Model (FTM) considers the fibers as a structure with an ideal round cross-section and a predefined, invariable fiber length [28]. The FTM can be used to simulate the fiber orientations; however, it has been proven that the kinematics of the fiber calculated with the FTM is faster than that of the real fiber and the overall fiber alignment is stronger [41]. According to the model, the fibers reach their stationary phase 2 to 10 times faster than reality [37]. Another inaccuracy of the FTM is the assumption of the IRD-term that random interactions occur between fibers, which means that the fiber interaction can be assumed to be the same in any spatial direction [40]. However, these random interactions are implausible at high fiber volume contents and fiber lengths [40,42].

There are many further developments of the FTM with the aim of improving the calculation of fiber orientations. As an example, the reduced strain closure (RSC) model can be cited. This model, introduced by Wang [37], calculates slower orientation kinetics than the FTM, but similar steady-state orientation [37,41]. Besides the RSC model, many other models can be mentioned, such as the Retarding Principal Rate (RPR) model, the Improved Anisotropic Rotary Diffusion Retarding Principal Rate (iARD-RPR) model, the Discrete Element Method (DEM), the Smoothed Particle Hydrodynamics (SPH) method [43], and the Moving Particle Semi-implicit method (MPS) [34].

This study focuses on the Moldflow^®^ Rotational Diffusion (MRD) model, which is the latest advancement of fiber orientation models, implemented in the commercial injection molding software Autodesk Moldflow^®^.

Like the FTM, the MRD model developed by Bakharev [44,45] is based on Jeffrey’s algorithms for translation and rotation [35,38], where $\;\dot{A}$ is the instantaneous time-dependent change of the fiber orientation tensor A. The algorithms consist of the velocity gradient tensor L (Equation (3)) with its component from Equation (4), where $v_{i}$ is the velocity component in the $x_{i}$ direction. W is defined as the vorticity tensor (Equation (5)), and D is defined as the rate-of-deformation tensor (Equation (6)). ξ considers the fiber aspect ratio, whereas l the fiber length and d the fiber diameter is. $A_{4}$ is the fourth-order orientation tensor, which must be calculated using the closure approximations like hybrid or orthotropic [45,46].
(1)$${\;\dot{A}=\dot{A}}^{\mathrm{Jeffrey}}{+\dot{A}}^{\mathrm{ARD}}$$
(2)$${\;\dot{A}}^{\mathrm{Jeffrey}}=\left(W\cdot A-A\cdot W\right)+\xi \left(D\cdot A+A\cdot {D-2A}_{4}:D\right)$$
(3)L=∇v=W+D
(4)$$L_{\mathrm{ij}}=\nabla_{j}v_{i}$$
(5)$$W=\frac{1}{2}{(L-L}^{T})$$
(6)$$D=\frac{1}{2}{(L+L}^{T})$$
(7)$$\xi =\frac{{\left(\frac{l}{d}\right)}^{2}-1}{{\left(\frac{l}{d}\right)}^{2}+1}$$

The difference between the FTM and MRD model is that the IRD term of the FTM has been replaced by a Moldflow^®^ specific anisotropic rotational diffusion (ARD) term to provide a better prediction of fiber orientation than the FTM [44]. The ARD term is shown in Equation (8), where C_i_ is the fiber interaction coefficient and $\dot{\gamma }$ is the strain rate defined in Equation (9). The MRD model adds 3 new parameters D_1_, D_2_, and D_3_, which serve as anisotropy factor tensor (see Equation (10)) [45]. Bakharev refers to the parameter values D_1_ = 1, D_2_ = 0.5, and D_3_ = 0.3 as optimal [44]. These parameters are implemented in Moldflow^®^ as default values.
(8)$${\;\dot{A}}^{\mathrm{ARD}}{=2C}_{i}\;\dot{\gamma }\left(\;\hat{D}\;-\mathrm{tr}(\;\hat{D}\;)A\right)$$
(9)$${\;\dot{\gamma }=(2D\;:D)}^{\frac{1}{2}}$$
(10)$$\hat{D}=\begin{array}{ccc} D_{1} & 0 & 0\\ 0 & D_{2} & 0\\ 0 & 0 & D_{3} \end{array}$$

The differential equation of the MRD model is solved by approximation through a system of algebraic equations using the Finite Element Method. For this, the specimen geometry must be discretized, which is done in Moldflow^®^ by generating midplane, dual-domain, and 3D meshes. The algorithm maps the CAD geometry with tetrahedra (3D) or triangles (midplane, dual-domain) [47]. In addition to the model parameters, the simulations are carried out with different global edge lengths (GEL) and resulting mesh densities to determine the influence of the discretization elements on the simulated fiber orientations and to find a mesh that allows calculating realistic results within an acceptable time.

To solve the algebraic system of equations, iterative methods are used. In this method, the solution vector is iteratively updated until the result satisfies the specified convergence criteria. Commercial CFD software packages have functions to monitor numerical methods for convergence solutions [48].

In this study, the influences of the parameters of the injection molding process and the IMFS on the real and simulated fiber orientation are presented. The real fiber orientation is characterized by means of high-resolution X-ray Microtomography. On the other hand, the influences of the parameters (C_1_, D_1_, D_2_, D_3_) of the simulation models (MRD) as well as different mesh parameters and densities for the discretization of the specimen are worked out. The aim is to optimize the model parameters in such way that the real fiber orientation can be represented as accurately as possible by the simulation.

## 2. Material and Methods

### 2.1. Used Material

For the manufacturing of test specimens, a polyamide 6 (PA6) with a glass fiber content of 30 wt.% (Ultramid B3EG6), provided by BASF (Ludwigshafen am Rhein, Germany), was used. According to the manufacturer, this material for injection molding processing offers high mechanical strength, stiffness, and thermal stability at a density of 1.36 g/cm^3^. It is therefore particularly suitable for components and machine elements as well as high-grade electrical insulation. The melting temperature (T_m_) of the material is 220 °C, and the melt volume rate (MVR) is 35 cm^3^/10 min (275 °C/5 kg) according to ISO 1133. The glass fibers show a diameter of approx. $d_{f}$ = 12 µm and a mean glass fiber length of approx. $l_{f}$ = 400 µm.

### 2.2. Investigated Specimen

The test specimen (see Figure 2) with the dimensions 4 mm × 10 mm × 80.4 mm was manufactured via injection molding process. The specimen shown is filled from the narrow head side via a film gate. In addition, the test specimen has an elongated opening (notch) in the center with the dimensions 4 mm × 2 mm with a radius of 1 mm over the entire specimen thickness of 4 mm. The notch represents an obstacle to the flowing melt in the injection molding process, resulting in melt deceleration and acceleration, which is reflected in a characteristic fiber orientation that can be observed with the µCT and the IMFS (see region 2 in Figure 2).

### 2.3. Injection Molding

Before processing by injection molding, the material was dried using an air dryer “Dry Jet Easy” from GfK (Igensdorf, Germany) until the residual moisture content of approx. 0.2% was achieved. The moisture was measured with the measuring instrument MA100Q from Sartorius (Göttingen, Germany).

A hydraulic injection molding machine “Allrounder 320 C (Golden Edition)” from the company Arburg (Loßburg, Germany) with a screw diameter of 25 mm, nozzle diameter of 2.5 mm, and a clamping force of 500 kN was used to carry out the injection molding process. Furthermore, a needle valve nozzle and a cold runner system were used.

To determine a robust operating point, the setting parameters of the injection molding machine were varied iteratively during the sampling process until the standard deviation of the part weight of the manufactured specimens was minimal. The determined machine parametersare referred to in the following as optimum process parameters (Table 1). The weight of the specimens was measured directly after manufacturing using the “Entris II” laboratory weighing scale from Sartorius (Göttingen, Germany).

According to the theory, the flow rate—derived from the translatory movement of the screw—and the melt temperature—controlled by the temperatures of the heating tapes—in the injection molding process have a significant influence on the resulting fiber orientation in the test specimens. To be able to investigate the influence of these parameters on the real fiber orientation in the injection-molded specimen as well as the simulated fiber orientation in the IMFS, the nozzle temperature and the flow rate were varied using a full factorial experimental design with a central point. The process parameter combinations resulting from the experimental design are listed in Table 2. For the parameters not listed in Table 2, such as the packing pressure profile, the values listed in Table 1 were used constantly. The sample name is derived from the values of nozzle temperature, mold temperature, and flow rate.

### 2.4. X-ray Microtomography Analysis of the Process Parameter Dependence of the Fiber Orientations

In order to characterize possible deviations of the fiber orientation from different injection molding parameter configurations, X-ray microtomography (µCT) measurements were carried out. These were realized on specimens of parameter configurations 1 and 9 to cover the largest difference in properties.

To enable a highly detailed analysis of the molding parameter-dependent fiber orientation and fiber length distribution, an additional characterization was conducted using an X-ray microscope (Zeiss Xradia Versa 520, Carl Zeiss, Oberkochen, Germany). The measurements were performed at a voltage of 80 kV and a current of 87 μA using the low energy filter LE2 and a magnification of 4. A total of 1601 images were acquired with a voxel size of 2.64 μm, binning setting 1, and an exposure time of 8 s for each image. The single images were reconstructed using the Zeiss XMReconstructor software.

Further image processing was carried out utilizing a 3D data visualization and analysis software system and the XFiber extension (Avizo 9.4, Thermo Fisher Scientific, Waltham, MA, USA) for the quantitative analysis of fiber properties. Cylinder correlation was used to characterize the fiber orientation in the specimens. To segment the fibers, a tracing algorithm was applied to the resulting correlation lines. The detected fibers can be quantitatively evaluated with regard to their length and orientation in the test specimen. XFiber extension settings for cylinder correlation and the tracing algorithm settings are shown in Table 3.

The performed µCT investigations proved that the fiber orientations in the existing sample geometry are symmetrical. Therefore, only halves of the specimen were investigated with the µCT, which improved the image resolution and accuracy. The µCT provides images in which the individual fibers can be detected due to gray level differences.

Three parameter settings were selected from the experimental design in Table 2 for characterization with the µCT: Setting 1, setting 5, and setting 9. Setting 5 is considered optimal based on the sampling process and would be used in the regular operation of the machine. With setting 1, the least differences in the velocity field are expected due to the lowest flow rate. In addition, the low melt temperature results in a high viscosity, which is considered flow resistance. In contrast, setting 9 represents the combination of the highest flow rate and nozzle temperature considered.

In order to characterize not only the influence of the injection molding parameters but also the influence of different flow situations in the test specimen on the fiber orientation, both the µCT investigations and the IMFS are carried out in two regions (region 1 and 2 in Figure 2) of the test specimen. In region 1, the melt exits the film gate, which slows down the melt. The gate can be interpreted as a divergent channel. In region 2, the melt flows in a simple shear flow until it is partially stopped by the notch and partially diverted into taper with a smaller cross-section. The taper can be interpreted as a convergent channel.

In order to show the occurring effects in region 2 at a higher resolution, this region is divided into 3 locations: simple shear flow (location 2.1), melting deceleration (location 2.2), and melting acceleration (location 2.3). These locations are shown in Figure 3. A qualitative comparison of the process-related fiber orientations in regions 1 and 2 is performed with partial volumes of the µCT images along the main planes (Figure 4).

For this purpose, one sample of simple shear flow (location 2.1), melt retardation (location 2.2), and melt acceleration (location 2.3) is considered (see Figure 4). From the samples, the orientations of the individual fibers are determined using the XFiber software described above. Individual fiber orientations are approximated into curves by moving averages over 20 periods and compared with each other to identify possible differences in fiber orientations for different process parameters.

### 2.5. Injection Molding Simulations for Prediction of Fiber Orientation

The simulations were performed with the software Autodesk Moldflow^®^ Insight 2019, version FCS 40.0.26.0. Moldflow^®^ uses the Navier–Stokes equations to simulate the flow. The viscosity is calculated with the Cross-WLF (Williams, Landel, Ferry) model and the pVT behavior with the 2-domain Tait model [47].

The corresponding material parameters of Ultramid B3EG6 are taken from the Moldflow^®^ database. The information about the injection molding machine (type: Arburg 320 C Golden Edition; manufacturer: Arburg, Loßburg, Germany) and the mold are taken from the respective datasheets. To be able to compare the results of the fiber orientation in the injection molded test specimens and the simulation as well as possible, the parameters for the simulation are adapted as closely as possible to the real injection molding process from Section 2.3. Equivalent to the real process, the simulation switches to holding pressure at a cavity filling level of 99% and displays the pressure profile from Table 1. After 20 s of cooling, the simulation is terminated with the virtual demolding of the specimen. The mold opening time is set to 9 s.

For the simulation of the fiber orientation, the Moldflow^®^ standard closure approximation called “Orthotropic 3” is used. All parameters mentioned so far remain constant in all simulations. The varied parameters are listed below in the respective sections.

### 2.6. Discretisation of the CAD-Model with 3D and Dual Domain Mesh

To solve the differential equation of the simulation model, the specimen geometry has to be discretized. The discretization is carried out by representing the entire sample geometry by quadrilateral elements (tetrahedron) in the case of a 3D mesh, or the surfaces of the sample by triangular elements with 3 nodes, triangular elements (triangle) in the case of a Dual-Domain mesh. Tetrahedrons and triangles are created in Moldflow^®^ with 3 parameters: global edge length, chord angle, and minimum number of tetra layers (only in the case of the 3D mesh).

The results of the simulations are related to the mesh density (number of mesh elements). As the global edge length (GEL) decreases, the mesh density increases due to the growing number of mesh elements, which increases the computation times. It is partly unclear with which mesh type and mesh parameters the best possible representation of the fiber orientation can be achieved with acceptable computation times. According to the theory, the simulation results approach the results of the real fiber orientation (characterized with the µCT) with increasing mesh density. The convergence criteria of the numerical solution for the filling and packing phases are shown in Table 4.

Table 5 shows the mesh parameters used in the study. The comparison in the case of the 3D mesh is performed in region 2 at location 2.1 and along the XZ and YZ planes. Since the Dual Domain mesh cannot show qualitative trajectories, locations 2.2 and 2.3 are examined quantitatively as a substitute to evaluate the prediction of fiber orientation at the notch and in the taper. The compared planes and locations are shown in Figure 5. Setting 5 from Table 2 is selected for the simulation. The MRD parameters are mesh-dependent default values from Moldflow^®^ for 3D and Dual Domain meshes. The general process and simulation settings are the same as in the configuration described in Section 2.3 and Section 2.5.

The results characterize the ability of the models to reproduce the shell-core layer effect and the fiber orientations in individual layers by fitting to the µCT results. In addition, the ability to fit the simulation results to the µCT results by varying the GEL is characterized.

### 2.7. Dynamic Image Analysis

One of the parameters of the MRD model is the aspect ratio of the fibers, which is calculated as the quotient of the fibers’ length and diameter (l/d). The aspect ratio of the fibers is assumed by Moldflow^®^ to have a constant value of 25. The data sheets of Ultramid B3EG6 do not specify the aspect ratio of the fibers used, so it is impossible to compare the aspect ratio values of Moldflow^®^ with the aspect ratio of the real fibers. It is also known from theory that fiber breaks occur during the injection molding process, which changes the length of the fibers and thus the aspect ratio. The aspect ratio of the fibers and their change during injection is to be determined experimentally for the simulations using dynamic image analysis.

In initial experiments with dynamic image analysis, the influences of the injection molding process on the fiber length distribution are investigated. According to Osswald [8], a large fiber shortening occurs, especially at low melt temperatures and high volume flow rates, due to high shear stresses. Therefore, setting 3 from the experimental design in Table 2 is chosen to study the fiber breakage. To show the process influence on fiber shortening, the fiber lengths are measured in the raw material, in the sprue, and at the end of the specimen (blue mark in Figure 2). In further measurements with dynamic image analysis, the fiber lengths are measured in regions 1 and 2 (red marking in Figure 2), in which the simulation results are later compared with the µCT-values. Specimens manufactured with setting 5 are used for this purpose since this setting is also used for the simulations.

The resulting fiber lengths in the samples were measured using the dynamic image analysis system QicPic (Sympatec, Clausthal-Zellerfeld, Germany) with a Mixcel liquid dispersion unit. The fibers of the above-mentioned regions of the specimens were separated from the matrix polymer according to ISO 1172 by burning the matrix in a muffle furnace from the company Ireos. Afterward, the fibers were dispersed in isopropanol and filled into the liquid dispersion unit of the measuring system, which provided a constant flow of the dispersed fibers.

This isopropanol flow with the fibers passed a cuvette with a thickness of 1 mm and a window at which a high-speed camera captured images of each fiber. Subsequently, the software Windox calculated the length and diameter of the fibers. The objective M7 with a resolution of 4.2 mm was employed, which realizes a minimal fiber size of 13.0 µm and a maximum fiber size of 8.66 mm. The values of the median, the 10th, and the 90th percentile were analyzed to acquire information about the scope of the distribution.

### 2.8. Determination of the Fiber Interaction Coefficient C_i_

The second parameter of the MRD model is the fiber interaction coefficient (C_i_), which is specific to each fiber matrix configuration. According to Ferec [39], a high C_i_-value leads to an isotropic prediction of the fiber orientation, while a low C_i_-value aligns the fibers in the flow direction and results in an anisotropic prediction. Due to the wide range of possible results by varying C_i_, it should be possible to obtain simulation results that reproduce the real fiber orientations from µCT well.

For this purpose, 5 simulations with different C_i_-values (see Table 6) are performed, comparing the results of the predicted fiber orientation in locations 2.1, 2.2, and 2.3 (see Figure 3 and Figure 4) with the µCT images at the corresponding regions. The fiber interaction coefficient in the simulation that best represents the real results in all three locations is interpreted as specific C_i_ of the material. The fiber interaction coefficient C_i_ varies in different sources. Moldflow^®^ assumes a value of C_i_ = 0.0015, while Osswald indicates a value of C_i_ = 0.03–0.06 for 30% glass fiber reinforced material [8].

Setting 5 (see Table 2) is chosen for the simulation. The general procedure and simulation settings are the same as described in Section 2.3 and Section 2.5, while the 3D mesh of Section 3.2 is used.

The experimental results are intended to show the ability of the MRD model to map the shell and core layers with different C_i_-values and to test the possibility of finding a simulation result that agrees with the µCT result.

### 2.9. Investigation of the D-Parameters

The remaining 3 parameters of the MRD model, D_1_, D_2_, and D_3_, are model specific and have an unknown effect on the simulation of the fiber orientations. The main purpose of investigating the D-parameters is to test the possibility of generating the core layer without compromising the good prediction of the shell layer.

Twelve simulations with different values for the D-parameters (see Table 7) are performed, varying each of the D-parameters individually. The default values of the D-parameters specified by Moldflow^®^ are the starting point for the variation, where the simulation results should map the µCT-curve best. Simulations with different D-values should produce deviations from the “optimal” results, but possibly represent the formation of the core layer. For the simulation, setting 5 from Table 2 is selected again. The general process and simulation settings correspond to the configuration mentioned in Section 2.3 and Section 2.5. In addition, the 3D mesh from Section 3.2 is used, while the C_i_-value determined in Section 3.6 is adopted for the variation of the D-parameters.

## 3. Results and Discussion

### 3.1. Influence of the Process Parameters on the Fiber Orientation

Figure 6 shows the results of a qualitative comparison of fiber orientation from µCT images taken in region 1 along the XZ and YZ planes. The presented images result from the specimens produced by different nozzle temperatures and flow rates, which were assumed to be significant for the formation of the fiber orientation. The red color indicates the fibers oriented in the flow direction. Blue color represents the fibers oriented transversely to the flow direction.

In region 1 (see Figure 2), the melt flows from the film gate with a thickness of 2 mm into the specimen with a thickness of 4 mm, corresponding to a divergent channel. After leaving the film gate (top side), part of the melt flow should rotate 90 degrees due to the increased specimen thickness according to Jeffrey’s model, which should lead to an immediate reorientation of the fibers transverse to the flow direction on the bottom side. Behind the film gate, however, this effect does not happen. The fibers in the middle of the film gate are almost all oriented in the flow direction (stationary phase). However, the fibers are reoriented transverse to the flow direction depending on the distance from the film gate, creating the core layer.

The most pronounced core layer can be seen in setting 1, which has the lowest flow rate due to the process parameters (265 °C, 30 cm^3^/s). The low melting velocity suggests that the velocity gradients, and hence the shear rates, should be lowest in laminar flow. At low shear rates, the zones dominated by the shear flow shrink and promote the elongational flow that causes the formation of the core layer, which is classical for divergent channels. However, the explanation is not compatible with the theory of the velocity field of the stationary phase of the fibers. Despite the differences in melt temperature and flow rate, the results of fiber orientation in settings 5 (280 °C, 45 cm^3^/s) and 9 (295 °C, 60 cm^3^/s) can be considered equivalent.

Figure 7 shows the results of a qualitative comparison of fiber orientation from µCT images taken in region 2 along the XZ and YZ planes.

As described in Figure 3, region 2 is divided into 3 locations. In the location of simple shear flow (2.1), the fibers are oriented in the core and shell layer. The layers can be interpreted as a logical continuation of the orientation formation from region 1. In the middle of the core layer, there is a variation between settings 1 and 9 (lowest and highest melting rates) that makes it difficult to compare the fiber orientation of the samples. The notch causes part of the fibers to decelerate, resulting in the disappearance of the shell-core layer effect and a complete reorientation of the fibers transverse to the flow direction. In the case of melt deceleration, the front fibers are decelerated immediately. The difference in velocity leads to fiber rotation transverse to the flow direction and to the formation of buckling stresses in the fiber. The remaining fibers flow into the notch with a smaller cross-section (converging channel), which accelerates the melt and causes the fibers to orient themselves completely in the flow direction (core layer disappears). Apart from the variation in the simple shear flow in settings 1 and 9, there are no significant differences between the specimen prepared with different process parameters.

As described in Section 2.4, curves of the moving average were derived from the individual values of the fiber orientation. The coordinate system describing the indices of the tensors (e.g., T_zz_) can be found in Figure 2. From this, it follows that the spatial direction component T_zz_ indicates the direction of the flowing melt, plotted on the Y-axis. The representation of the tensor values and their meaning are shown in Figure 2. From these figures, it follows that fibers with a T_zz_-values close to 1 are oriented in the flow direction. The fiber orientation is plotted over the specimen thickness of 4 mm. Figure 8 shows a comparison of these curves for varying process parameters in the single shear flow in location 2.1.

The representation of the fiber orientation over the specimen cross-section shows that the shell layer is very similar for the shown settings 1, 5, and 9. This can be seen from the overlaid curves in the range of 0–1000 µm and 3000–4000 µm, i.e., the edge regions of the specimen. It can be concluded that the flow rates and temperatures considered for settings 1, 5, and 9 have no significant effect on the formation of the shell layer.

The core layer, in the range 1500–2500 µm, is more pronounced in setting 5 than in settings 1 and 9, as indicated by the low values in the T_zz_ direction. This means that in setting 5, there are fewer fibers oriented approximately in the flow direction in the core layer than setting 1 (lowest shear rate) and setting 9 (highest shear rate).

At this point, it must be pointed out that it cannot be excluded that the differences in fiber orientations in the core layer are caused by the above-mentioned fluctuations and not by the changed process parameters. However, based on the results, it is assumed that the process parameters influence the formation of the core layer, while the shell layer can be considered very similar in all cases.

The shell layer presents itself very similarly for all process parameters used. In contrast, the variation of the process parameters leads to a change and fluctuations of the orientations in the core layer, which, however, are not significant.

Regarding these fluctuations, the evaluation mechanism has to be considered, where the single fibers detected by the µCT and their orientations are first merged to a point cloud plotted over the specimen thickness. These clouds are mathematically approximated to functions (here: moving average over 20 periods), which on the one hand leads to a better representation, but on the other hand also to a loss of valuable information.

### 3.2. Discretization of the CAD-Modell—3D Mesh

For the 3D mesh, the specimen is completely mapped with the global edge length of 1 mm and 0.5 mm. A combined mesh is used to map the test specimen with a global edge length of 0.25 mm. This combined mesh only maps the examined regions with the global edge length of 0.25 mm, while the remaining subregions are mapped with a coarser mesh (here: 0.5 mm and 0.75 mm) to reduce the computation time.

Meshing with a global edge length of 1 mm results in a mesh with 2,056,989 elements, an edge length of 0.5 mm results in 6,822,455 elements, and an edge length of 0.25 mm results in 5,488,354 elements.

Figure 9 and Figure 10 show the results of a qualitative comparison between the real fiber orientation characterized with the µCT results and the fiber orientation prediction results by the 3D mesh in region 2, varying the global edge length (GEL) acquired along the XZ and YZ planes. The legend at the bottom of the figures refers to the results of the simulations but can also be used to evaluate the µCT image.

The simulations with the smallest global edge length (GEL = 0.25 mm) and thus the highest mesh density can work well for both the orientations in the simple shear flow and in the acceleration of the melt in the taper. However, the simulation with the highest mesh density cannot reproduce the shell core layer effect because the core layer is missing. The reason for this effect could be the fact, already known from the Folgar Tucker Model, that the models assume the kinematics of the fibers 2 to 10 times faster than in reality. For a mesh with a small GEL, the differential equation of Moldflow ^®^ Rotational Diffusion Model (MRD) is solved more frequently than for a mesh with a large GEL. As a result, the simulated fibers reach the stationary phase much faster than the real fibers.

In any simulation, there is a fluctuation in the area of the taper. The cause of this fluctuation is attributed to the calculation of the solution of the differential equation by the Moldflow^®^ solver and is not explained in detail here.

Figure 11 shows the results of a quantitative comparison of the fiber orientation of simulation results and µCT in the flow direction across the specimen width in region 2 at location 2.1. Simulation results were calculated using a 3D mesh and varying global edge lengths (GEL).

It can be seen that the 3D mesh with a global edge length of 1 mm reproduces the shell-core layer effect best, which contradicts the theory of finite element methods [49]. The core layer becomes smaller, and the fibers are increasingly present in the T_zz_-direction as the global edge length decreases. In each of the simulations, the shell layer is at a similar level but above the real values characterized by the µCT. As the global edge length decreases (increasing mesh density), the curves become flatter and asymmetrical. None of the three meshes can adequately represent the shell-core layer.

For further investigation, a combined mesh, as shown in Figure 12, is used since the theory of Computational Fluid Dynamics (CFD) states that the best results are obtained with the lowest GEL (the highest mesh density). The correct representation of the core and shell layers can be predictably realized by future variations of the model parameters. The resulting mesh consists of 3,975,131 tetrahedrons.

### 3.3. Discretization of the CAD-Modell—Dual Domain Mesh

As described in Section 2.6, in addition to the 3D mesh, a 2D Dual Domain mesh is also used to simulate the fiber orientation.

Figure 13 shows the results of a quantitative comparison of the fiber orientation from the µCT with the simulation results calculated with the Dual Domain mesh. Here, region 2 at location 2.1 is considered, and the global edge length (GEL) of the Dual Domain mesh is varied (0.5, 0.86, and 1.0 mm). The results of the simulation with the final 3D mesh from Figure 12 and default MRD parameters are also listed for reference purposes.

Similar to the use of the 3D mesh, the smallest global edge length and thus the highest mesh density does not lead to the best results. Compared to the real fiber orientation from the µCT, the simulation with a global edge length of 0.86 mm specified by Moldflow^®^ provides the most accurate results and can best reproduce the shell-core layer effect in areas between 1000 µm and 3000 µm. In the edge regions (0–750 µm and 3250–4000 µm), the simulations show a rapid increase in fiber orientations in the T_zz_-direction. In the range of about 300 µm and 3700 µm, this is interrupted and the fiber orientation decreases rapidly. This effect is only partially visible on the µCT images.

As described in Section 2.6, the Dual Domain mesh maps only the surfaces of the specimen, so the specimen thickness must be calculated differently than in the case of a 3D mesh. Moldflow^®^ seems to have an algorithm that takes the dependence of the specimen thickness in the case of a Dual Domain mesh into account. For simple shear flows, simulation with Dual Domain mesh is superior to simulation with 3D mesh.

Figure 14 shows the results of a quantitative comparison of the simulation results calculated with the Dual Domain mesh varying the global edge length and the fiber orientation characterized with the µCT at locations 2.2 and 2.3.

In the case of melt acceleration and deceleration, the simulations with the Dual Domain mesh do not adequately represent the fiber orientations of the domains. The results of the simulations with the 3D mesh are clearly superior to the simulations with the Dual Domain mesh in this case.

Compared to the results of the µCT-analysis and the simulations with the 3D mesh, the simulations with the Dual Domain mesh reproduce the fiber orientations in a simple shear flow (except for the outermost edge regions) very well (Figure 13). Regarding the melt acceleration (location 2.3) and deceleration (location 2.2), the results of the simulations with Dual Domain mesh do not show good agreement (see Figure 14).

### 3.4. Dynamic Image Analysis

As described in Section 2.6, the process-related fiber length reduction is represented by measurements with dynamic image analysis.

In the first measurements with the dynamic image analysis, the influences of the injection molding process on the fiber length distribution are investigated. In order to demonstrate the expected significant fiber length reduction at a low melt temperature and a high shear rate, specimens of setting 3 (265_85_60) were characterized. To observe the entire flow path of the melt, fiber length measurements were performed on granules as well as on material from the sprue and the end of the specimen. In a second measurement, the fiber lengths of setting 5 (280_85_45) are measured in regions 1 and 2, in which the simulation results are later compared with the µCT-values (see Table 2).

The fibers in SGFRP are generally not present with uniform lengths but exhibit a specific fiber length distribution. In order to better compare these distributions and the underlying values, the values of the 10th, 50th, and 90th percentiles were derived from the distributions (see Table 8).

From the results of the fiber length measurement of the first measurement, a significant fiber shortening between the granules and the sprue can be observed. In contrast, the fiber length reduction between the sprue and the end of the specimen is not significant.

In the second measurement, differences in fiber lengths between granules and regions 1 and 2 of the specimen can also be seen. Here, the fiber length reduction between regions 1 and 2 is particularly noticeable, which is attributed to fiber breaks in the region of the notch. As described above, some of the fibers are stopped at the notch, and the remaining fibers are deflected into the taper and collide with the fibers flowing directly into the taper. The resulting deflections and collisions exert compressive and buckling stresses [28], which become more noticeable with increasing fiber length [8] and result in local fiber shortening.

### 3.5. Fibers Aspect Ratio Calculation

The aspect ratio (L/d) of the fibers is calculated from the quotient of the respective 50th percentile of fiber length (L) and diameter (d) (see Table 8) and is listed in Table 9.

Assuming that the fiber diameters are the same in each percentile, the differences in aspect ratio in different regions are due to the different fiber lengths. Based on the generated results, of which the value of 24.69 in region 2 is considered particularly representative, the expected value of 25 from the Moldflow^®^ database is adopted for the simulations.

### 3.6. Determination of the Fiber Interaction Coefficient Ci

After the influences of the process parameters and the discretization of the models, the influences of the model parameters on the simulated fiber orientation will be presented in this section.

Figure 15 shows the influence of the varied fiber interaction coefficient C_i_ on the results of the simulated fiber orientation in quantitative comparison to the results of the µCT-analysis using the example of the simple shear flow in location 2.1. The simulation results were calculated using a 3D mesh from Section 3.2.

As the value of C_i_ increases, the shell-core layer effect becomes more apparent in the simulation results, with none of the considered C_i_-values adequately representing the real fiber orientation. With low C_i_-values, the simulated fiber orientation curve shifts upward, and with high values of C_i_, it shifts downward, allowing the shell layer to fit the results of the µCT-analysis. In contrast, the core layer cannot be mapped by varying C_i_. Furthermore, the curve becomes asymmetric with decreasing C_i_-values. The simulation results for the shell layer as well as the core layer come closest to the results of the µCT-curve at a value of C_i_ = 0.1.

Figure 16 shows the results of a quantitative comparison of the µCT-analysis and simulated results calculated with the 3D mesh in region 2 at location 2.2, varying the fiber interaction coefficient C_i_.

In region 2, the fibers are reoriented transverse to the flow direction (low T_zz_-values). The core layer thus extends over almost the entire cross-section. In the very narrow edge regions, a layer with increased orientation in the $T_{\mathrm{zz}}$-direction is visible. All simulations can show the increased orientations in the edge regions, but cannot reproduce them exactly. The orientation of nearly all fibers in the core layer transverse to the flow direction can be reproduced by all simulations. As the C_i_-value decreases, the simulation results converge to the real fiber orientation, but in the middle of the specimen (at about 2000 µm), a zone of increased orientations in the T_zz_-direction emerges. The best results are obtained for simulations with 0.00001 < C_i_ < 0.1, which allows a fit to the µCT results, but also does not accurately map the edge regions.

Figure 17 shows the results of a quantitative comparison of the µCT results and the simulation results calculated with a 3D mesh and a varying fiber interaction coefficient C_i_ in region 2 at location 2.3 (melt acceleration).

The shell-core layer effect is not visible because the core layer disappears in location 2.3. Almost all fibers are aligned in the flow direction in both the simulations and the µCT-analysis (high-T_zz_ values). All considered values for C_i_ in the simulations can predict the remaining shell layer in the edge regions, with the most realistic results for a simulation with C_i_ = 0.01. Again, variation to higher C_i_-values shifts the curve downward so that the fit to the curve of the µCT-analysis is possible.

In a simple shear flow, the core layer cannot be mapped, and a variation of C_i_ can only represent the shell layer. By fibers deceleration, the classical shell core layer effect disappears. A small shell layer is visible in the edge regions, with the core layer dominating nearly the entire cross-section. By fiber acceleration, all fibers are oriented in the flow direction (disappearance of the core layer). The simulations can map the fiber orientation by varying C_i_ during fiber deceleration (location 2.2) and acceleration (location 2.3). C_i_ will be determined as 0.01 for coming D-investigations because C_i_ = 0.01 produces good results in all cases and corresponds nearly to the results from Osswald’s book [8].

### 3.7. Investigation of the D-Parameters

Finally, in this section, the influences of the D-parameters on the simulated fiber orientation will be presented.

Figure 18 shows the results of a quantitative comparison between the fiber orientation of the µCT-analysis and the simulation results calculated with the 3D mesh in region 2 at location 2.1 (simple shear flow) under variation of parameter D_1_.

In the present case, the variation of D_1_ has only a minor effect on the representation of the core and shell layers. The variation shifts the curve up (high D_1_) and down (low D_1_). Neither the shell nor the core layer is realistically represented by the variation of the parameter D_1_.

Figure 19 shows the results of a quantitative comparison of the real fiber orientation with simulation results through a 3D mesh and a variation of the parameters D_2_ and D_3_, in location 2.1 (simple shear flow).

Both the variation of the parameters D_2_ and D_3_ has no influence on the formation of the core and the shell layer in the present case. The variation of D_2_ shifts the curve upwards (high D_2_) and downwards (low D_2_), whereas the curves show an approximately equal course independent of the value of D_2_. None of the values used for D_2_ results in a curve of simulated fiber orientation that represents reality.

When varying D_3_, the curves are not shifted but show a change in their shape. With decreasing value of D_3_, the curves for fiber orientation show a more asymmetric course. Regardless of the value of the parameter D_3_, none of the simulated curves can adequately represent the real fiber orientation.

The results of the D-parameter variation do not meet the expectation. The optimal values for the D-parameters defined in theory do not provide the best results in the case of simulation with a 3D mesh. Depending on the interpretation, the simulations with values deviating from the “optimum” approach the µCT curve better than those with optimal values but are still inadequate because of missing core layer prediction.

The phenomenological nature of the model is primarily given as the reason for this discrepancy. It is known that all fiber orientation models are based on Jeffrey’s model. The differences between the models are due to different IRD or ARD terms. Jeffrey’s model was developed based on experiments with a particle in a Newtonian fluid, which has a different melt velocity profile than a viscoelastic plastic melt. In a Newtonian fluid, there are velocity differences between laminar layers along the entire cross-section flowed through. In a viscoelastic fluid, these differences disappear in the middle of the cross-section, especially in thick parts.

The Jeffrey-based models provide good predictions in a shear flow and in any laminar flow in which there are high-velocity differences between the individual laminar layers. This fact is of great importance for the calculation of the structural properties of, for example, a part, since they are based on the results of the injection molding simulations. The fact that the MRD model cannot represent the core layer in thick parts must always be taken into account by structural mechanics.

## 4. Conclusions

In this article, the fiber orientation in a 4 mm thick specimen made of short glass fiber reinforced polyamide 6 by using different parameter settings in injection molding was investigated by µCT-analysis. Further, the same specimen and the related fiber orientation were simulated using Moldflow^®^ simulation software with the MRD model. Results of µCT-investigation were compared with results from simulations regarding the fiber orientation in three locations of the specimen: simple shear flow, melt deceleration, and acceleration. The following aspects were investigated during the simulation:The influence of the process parameters on the resulting fiber orientations in injection molded specimens was investigated. It was found that the process parameters have only a minor influence on the formation of the shell layer with fiber orientations in the flow direction. In contrast, there is a significant effect of the process parameters on the formation of the core layer, in which the fibers are mainly oriented transverse to the flow direction.Regarding the influence of the mesh type (3D or Dual Domain) and the mesh density on the simulated fiber orientation, an effect of the global edge length was shown. In the case of solving the MRD differential equation, the finest mesh (highest mesh density) does not give the best results for either the 3D mesh or the Dual Domain mesh. The 3D mesh reproduces the fiber orientations in the region of melt deceleration and acceleration better than the Dual Domain mesh. In contrast, the Dual Domain mesh is superior in the region of simple shear flow.The aspect ratio of the fibers and the fiber length reduction in different areas of the specimen and granules was investigated. It was found that the fiber aspect ratio varied in the different areas due to process-induced fiber shortening. The variations are not significant, so the aspect ratio of 25 specified by Moldflow^®^ was used for the simulations.The simulation-based determination of the fiber interaction coefficient C_i_ and its influence on the resulting fiber orientation prediction showed that the coefficient C_i_ could be determined based on matching the shell layers in the simulations. However, it is not possible to find a value of the fiber interaction coefficient that adequately represents the core layer.The Influence of the MRD-specific model parameters D_1_, D_2,_ and D_3_ on the resulting fiber orientation prediction shows that it is not possible to adequately map the core layer by varying only one D parameter.

From the conclusions, the following scheme for performing the simulations can be derived:

Two simulations are required to adequately predict the fiber orientation. The first simulation is performed with a 3D mesh as dense as possible and MRD default parameters to represent the behavior of the fibers during melt redirection, melt acceleration (converging channel), and melt deceleration. The second simulation is performed with a Dual Domain mesh, a global edge length specified by Moldflow^®^, and MRD default parameters to represent the fiber orientation in a simple shear flow. The default geometric fiber values of Moldflow^®^ can be adopted for the two simulations as well as all other Moldflow^®^ default settings not mentioned here.

## Figures and Tables

**Figure 1 polymers-14-00029-f001:**
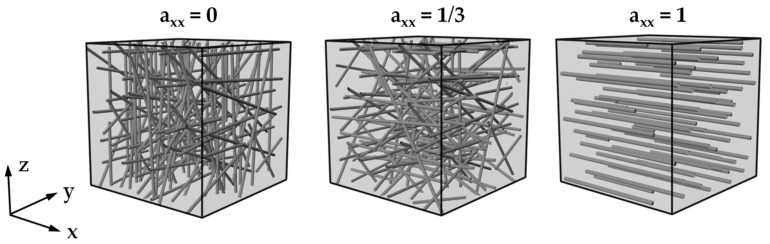
Evaluation of the fiber orientation distribution with tensor components using the example of the tensor a_xx_ with fibers oriented perpendicular to the corresponding axis (a_xx_ = 0), a three-dimensional random distribution (a_xx_ = 1/3), and fibers oriented in the direction of the corresponding axis (a_xx_ = 1).

**Figure 2 polymers-14-00029-f002:**
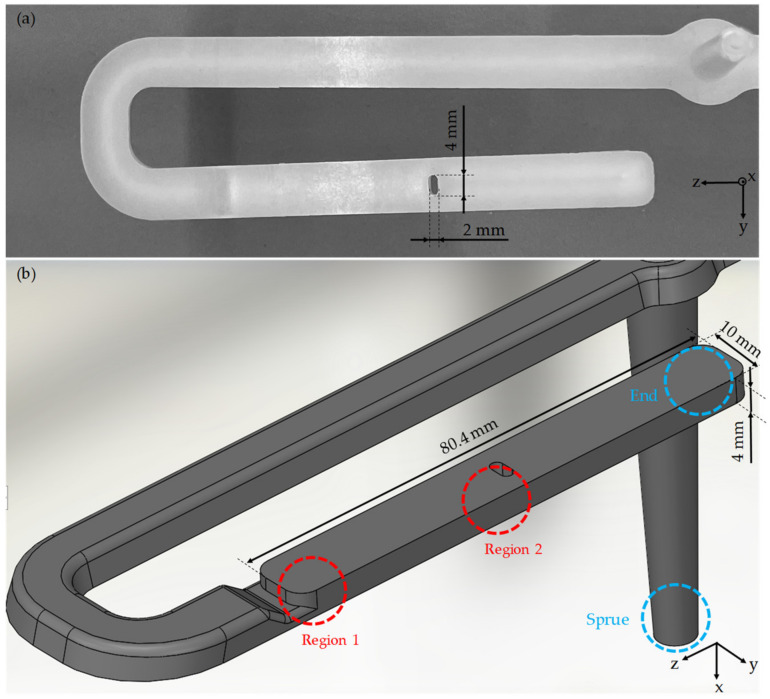
Photo (**a**) and CAD representation (**b**) of the manufactured test specimen with central notch and markings of the regions used for characterization and simulation.

**Figure 3 polymers-14-00029-f003:**
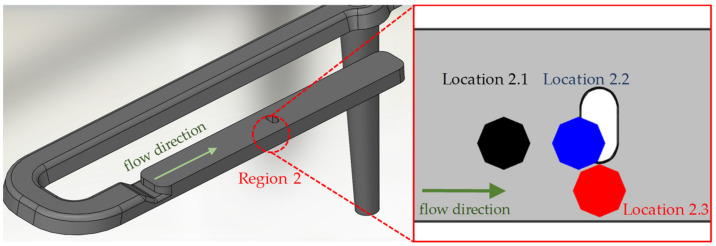
Positions of µCT-investigation in the area of the notch (region 2), subdivided into simple shear flow (location 2.1), melting retardation (location 2.2), and melting acceleration (location 2.3).

**Figure 4 polymers-14-00029-f004:**
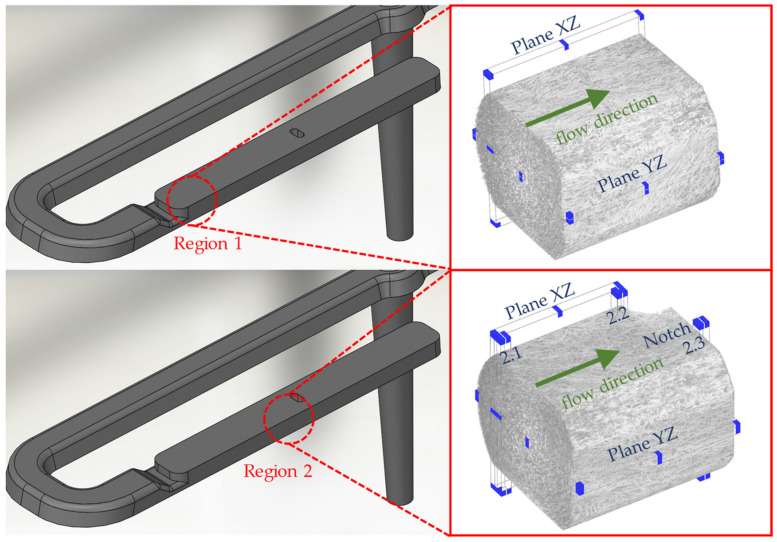
Raw data of 3D µCT images in regions 1 and 2 with underlying planes.

**Figure 5 polymers-14-00029-f005:**
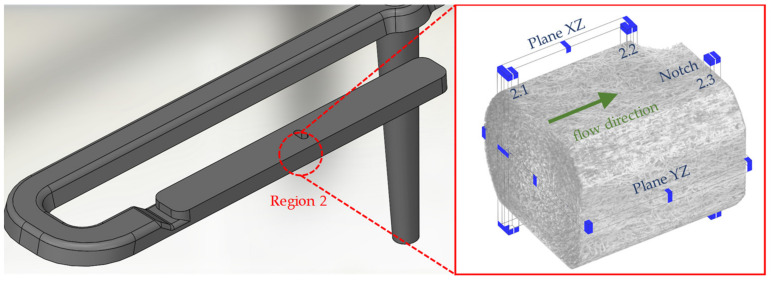
Region and planes investigated to evaluate the quality of the 3D discretization.

**Figure 6 polymers-14-00029-f006:**
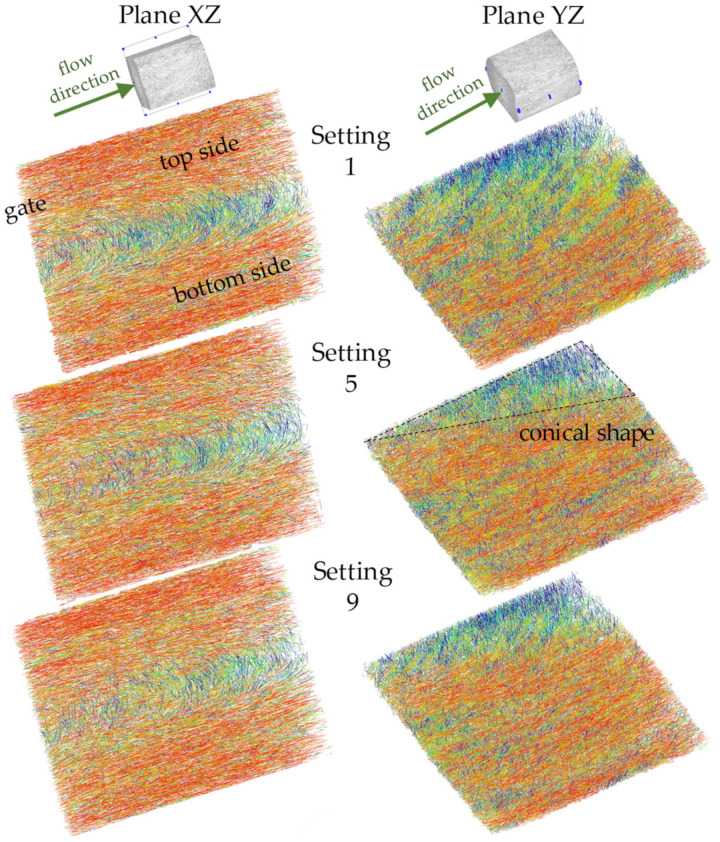
Qualitative comparison of fiber orientations from µCT analysis in the XZ- and YZ-plane of region 1 after specimen manufacturing with setting 1, 5, and 9.

**Figure 7 polymers-14-00029-f007:**
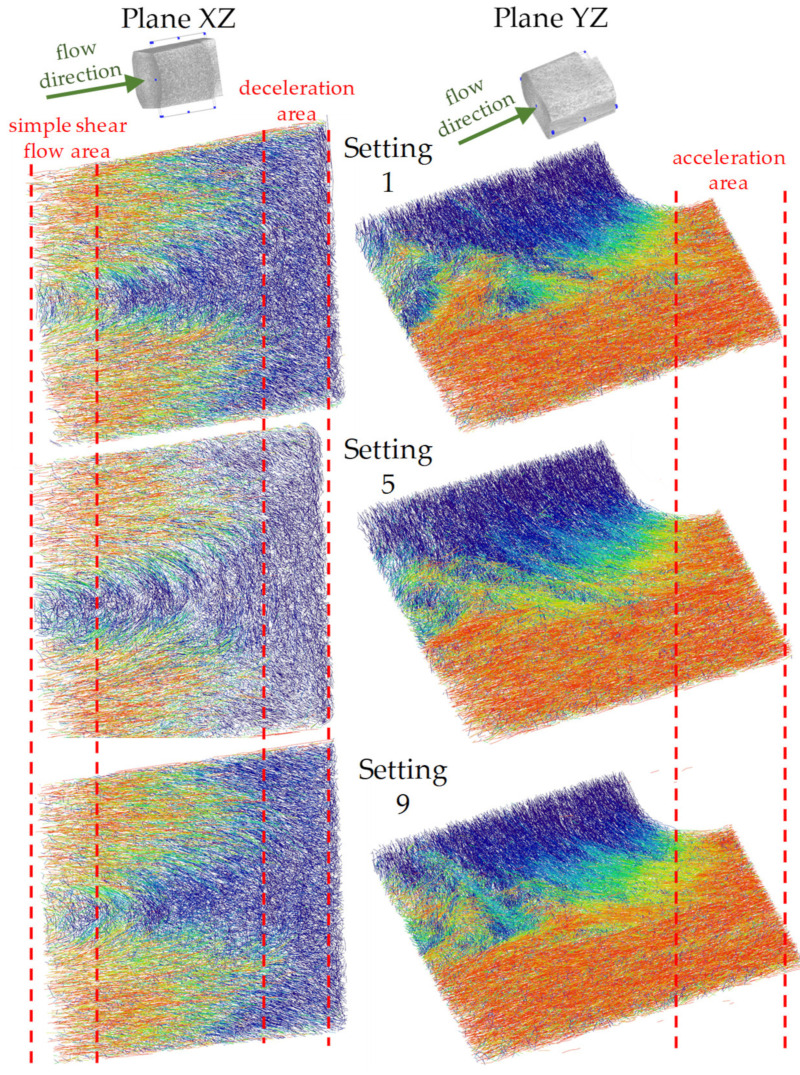
Qualitative comparison of fiber orientations from µCT analysis in the XZ- and YZ-plane of region 2 after specimen manufacturing with setting 1, 5, and 9.

**Figure 8 polymers-14-00029-f008:**
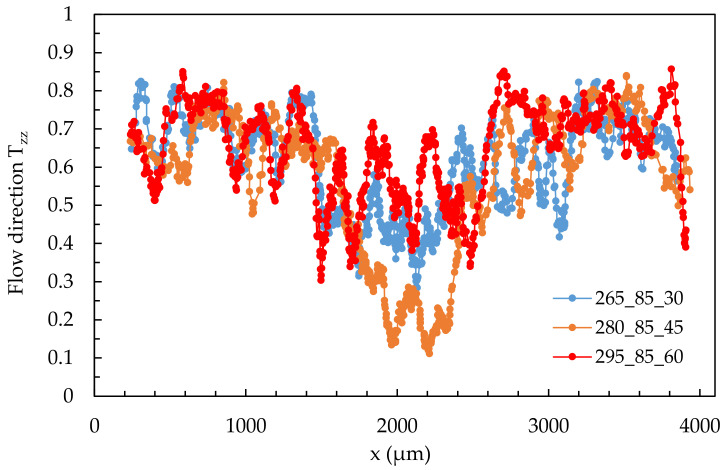
Quantitative results of fiber orientation in T_zz_-direction over the cross-section of specimen manufactured at process settings 1, 5, and at location 2.1.

**Figure 9 polymers-14-00029-f009:**
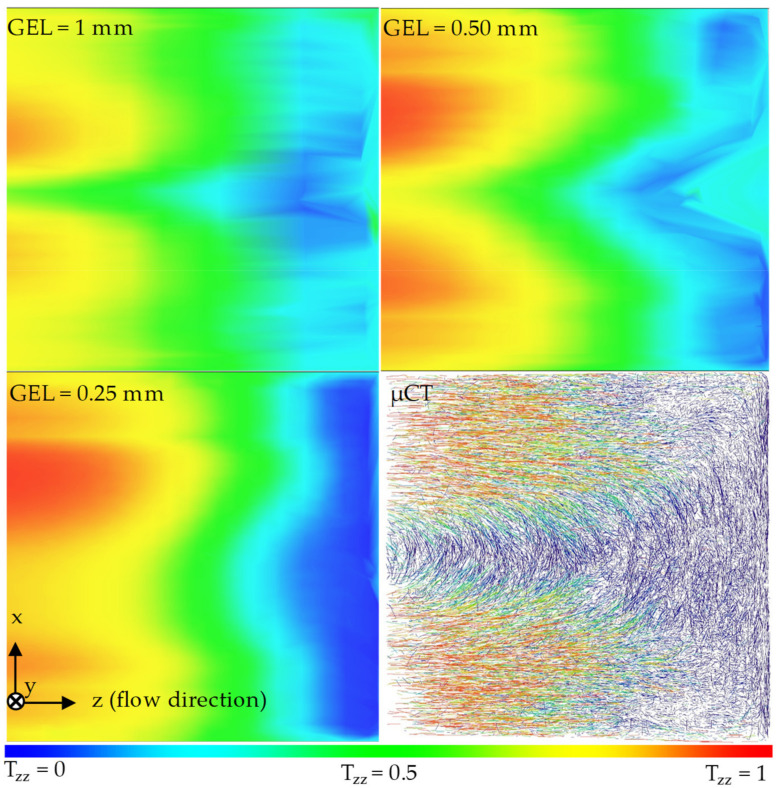
Qualitative comparison between the real fiber orientation characterized by µCT and the fiber orientation predicted by the 3D mesh in region 2 under variation of the GEL along the XZ plane.

**Figure 10 polymers-14-00029-f010:**
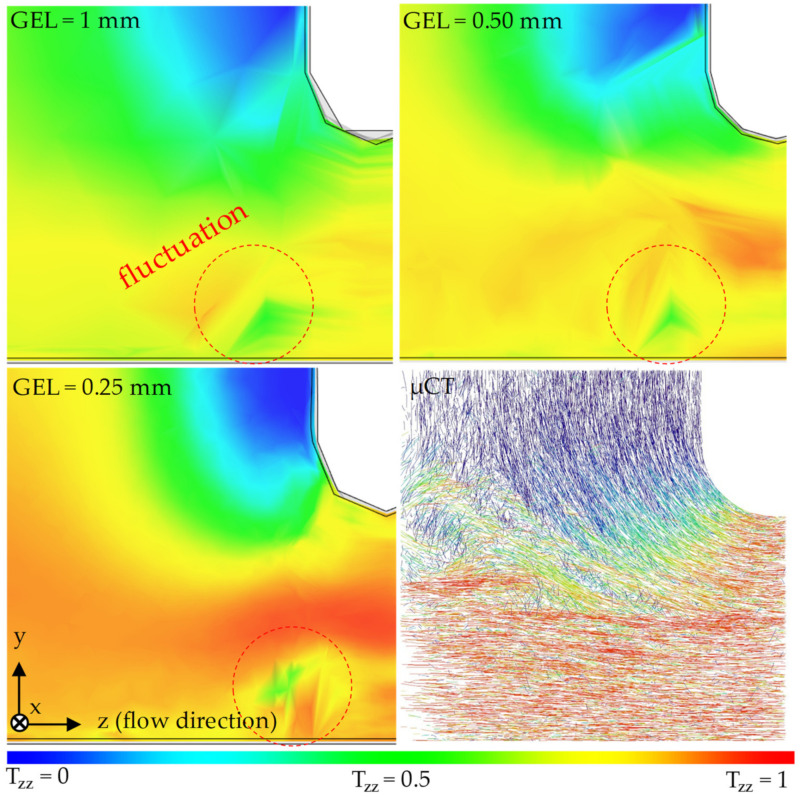
Qualitative comparison between the real fiber orientation characterized by µCT and the fiber orientation predicted by the 3D mesh in region 2 under variation of the GEL along the YZ-plane.

**Figure 11 polymers-14-00029-f011:**
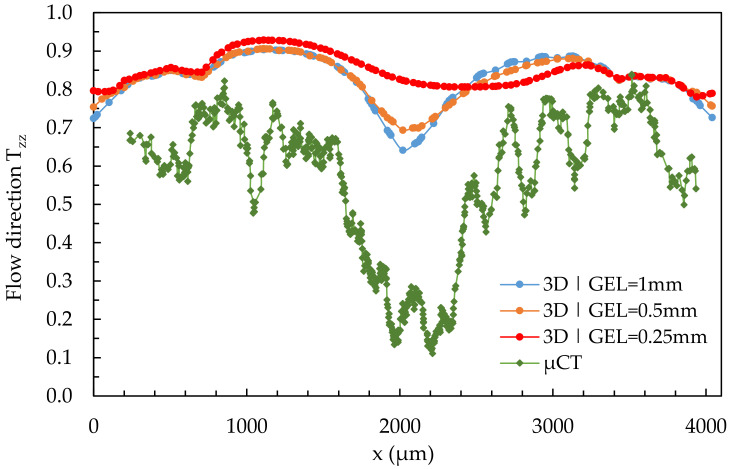
Quantitative results of fiber orientation in T_zz_-direction over the cross-section characterized by µCT and predicted by the 3D mesh in region 2 under variation of the GEL at location 2.1.

**Figure 12 polymers-14-00029-f012:**
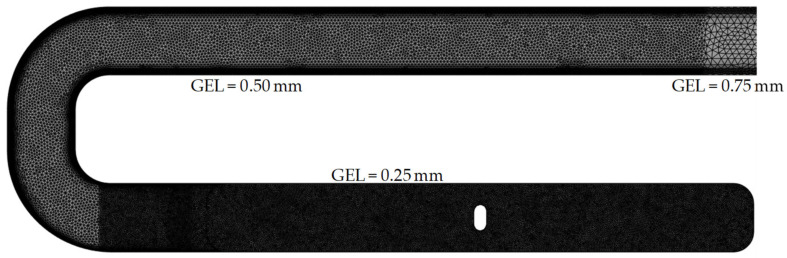
Final combined mesh with different GEL chosen for the MRD-parameter variation.

**Figure 13 polymers-14-00029-f013:**
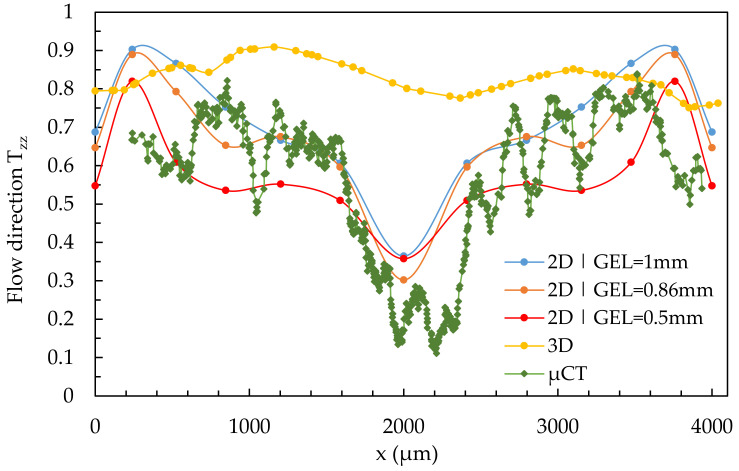
Quantitative results of fiber orientation in T_zz_-direction over the cross-section characterized by µCT and predicted by the Dual Domain mesh in region 2 under variation of the GEL at location 2.1.

**Figure 14 polymers-14-00029-f014:**
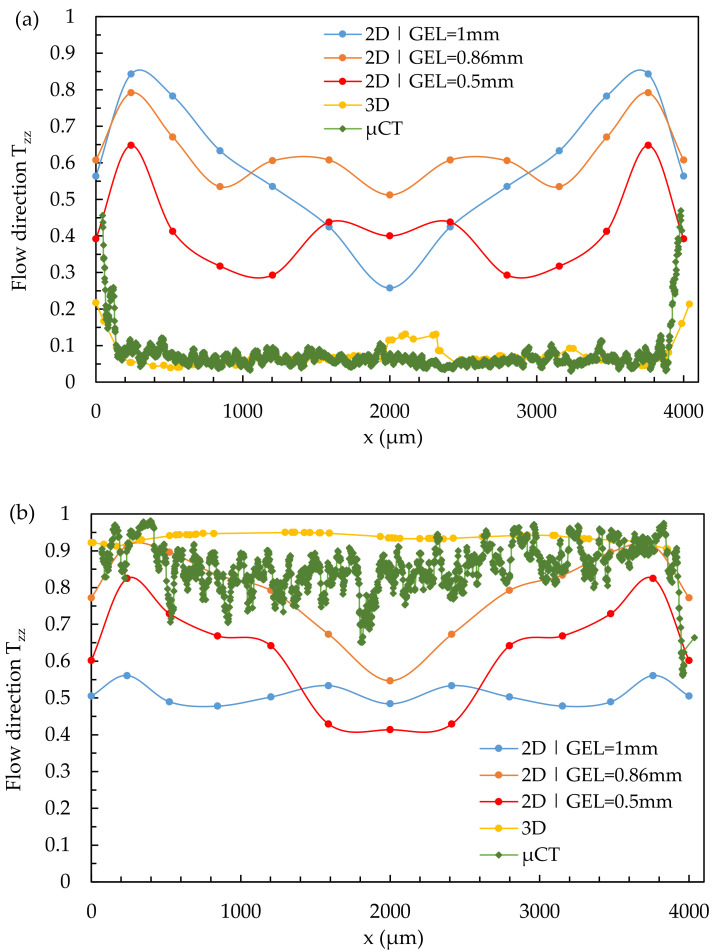
Quantitative results of fiber orientation in T_zz_-direction over the cross-section characterized by µCT and predicted by the Dual Domain mesh under variation of the GEL at locations 2.2 (**a**) and 2.3 (**b**).

**Figure 15 polymers-14-00029-f015:**
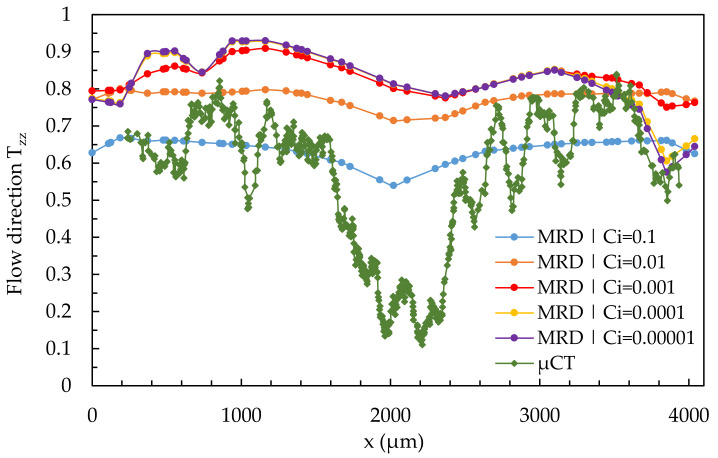
Quantitative results of fiber orientation in T_zz_-direction over the cross-section characterized by µCT and predicted by the 3D mesh under variation of the fiber interaction coefficient C_i_ at location 2.1.

**Figure 16 polymers-14-00029-f016:**
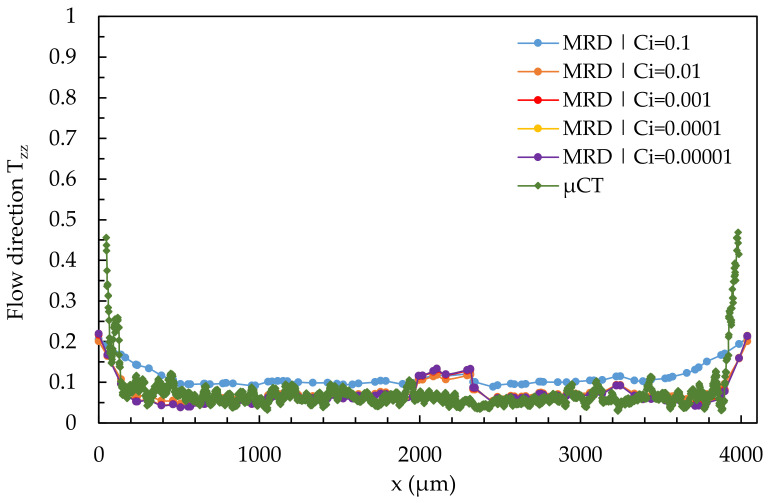
Quantitative results of fiber orientation in T_zz_-direction over the cross-section characterized by µCT and predicted by the 3D mesh under variation of the fiber interaction coefficient C_i_ at location 2.2.

**Figure 17 polymers-14-00029-f017:**
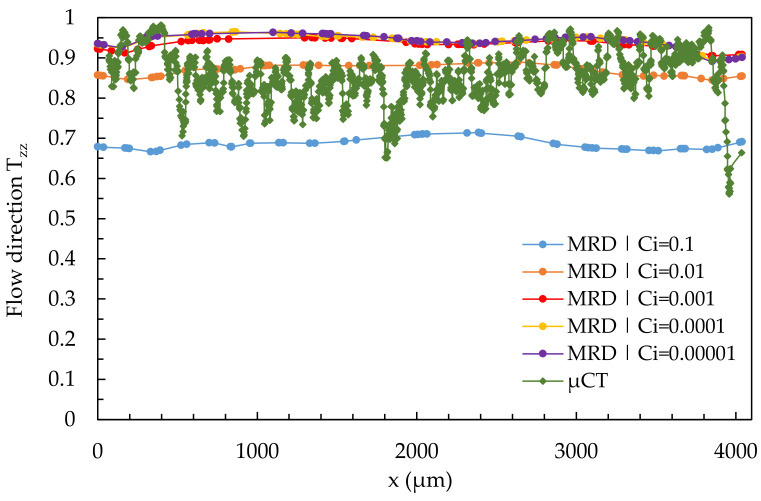
Quantitative results of fiber orientation in T_zz_-direction over the cross-section characterized by µCT and predicted by the 3D mesh under variation of the fiber interaction coefficient C_i_ at location 2.3.

**Figure 18 polymers-14-00029-f018:**
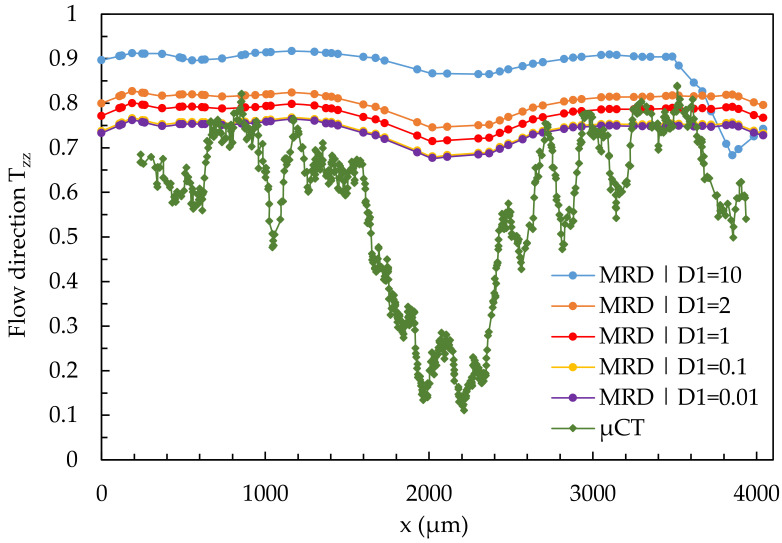
Quantitative results of fiber orientation in T_zz_-direction over the cross-section characterized by µCT and predicted by the 3D mesh under variation of the parameter D_1_ at location 2.1.

**Figure 19 polymers-14-00029-f019:**
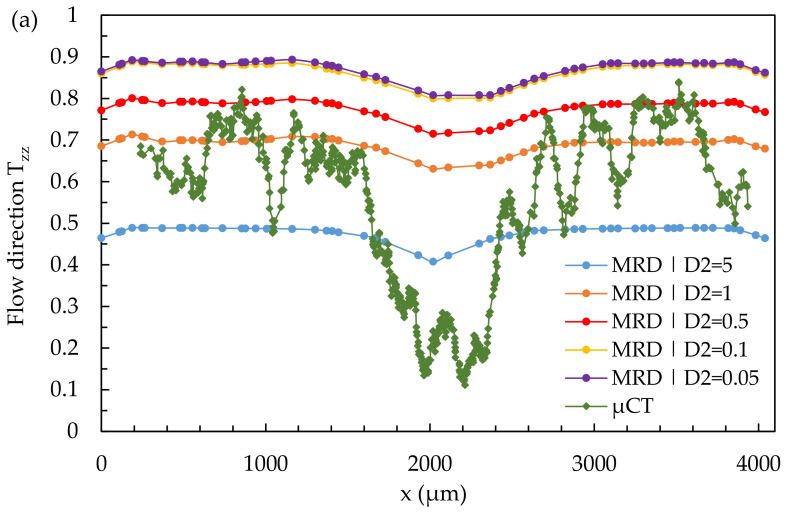
Quantitative results of fiber orientation in T_zz_-direction over the cross-section characterized by µCT and predicted by the 3D mesh under variation of the parameters D_2_ (**a**) and D_3_ (**b**) at location 2.1.

**Table 1 polymers-14-00029-t001:** Optimum parameters for the performed injection molding manufacturing process.

Parameter	Values
Nozzle temperature (°C)	280
Mold temperature (°C)	85
Packing pressure profile (MPa)	45.0; 45.0; 2.5
Packing pressure time steps (s)	0.05; 14; 0.05
Flow rate (cm^3^/s)	45
Dosing volume (cm^3^)	35
Switching volume (cm^3^)	15
Residual cooling time (s)	6

**Table 2 polymers-14-00029-t002:** Experimental design for the manufacturing of the test specimens by injection molding.

Setting No.	Name	Nozzle Temperature (°C)	Mold Temperature (°C)	Flow Rate (cm^3^/s)
1	(265_85_30)	265	85	30
2	(265_85_45)	265	85	45
3	(265_85_60)	265	85	60
4	(280_85_30)	280	85	30
5	(280_85_45)	280	85	45
6	(280_85_60)	280	85	60
7	(295_85_30)	295	85	30
8	(295_85_45)	295	85	45
9	(295_85_60)	295	85	60

**Table 3 polymers-14-00029-t003:** Used settings for fiber tracing with the XFiber extension (Avizo 9.4) based on the μCT data.

X-ray Microtomograph	Units	Zeiss Xradia Versa 520
Cylinder length	(μm)	38
Angular sampling		5
Mask cylinder radius	(μm)	10
Outer cylinder radius	(μm)	6
Minimum seed correlation		190
Minimum continuation quality		50
Direction coefficient		0.1
Minimum distance	(μm)	12
Minimum length	(μm)	38

**Table 4 polymers-14-00029-t004:** Convergence criteria for filling and packing phase.

Simulation Parameter		Filling Phase	Packing Phase
Maximum %volume to fill per time step	(%)	1	
Maximum packing time step	(s)		0.088
Iteration limit	(-)	50	50
Convergence tolerance factor (multiplies default)	(-)	1	1

**Table 5 polymers-14-00029-t005:** Simulation parameters.

Simulation Parameter		3D Mesh	Dual Domain Mesh
Chord angle	(°)	60	60
Minimum number of layers	(-)	40	-
Global edge length	(mm)	1; 0.5; 0.25	1; 0.86; 0.5

**Table 6 polymers-14-00029-t006:** Investigated values of the parameter C_i_ in the MRD model.

Test Point	C_i_	D_1_	D_2_	D_3_
1	0.1	1	0.5	0.3
2	0.01	1	0.5	0.3
3	0.001	1	0.5	0.3
4	0.0001	1	0.5	0.3
5	0.00001	1	0.5	0.3

**Table 7 polymers-14-00029-t007:** Investigated values of the D-parameters in the MRD model.

**D_1_ Investigation**				
Test point	C_i_	D_1_	D_2_	D_3_
1	0.01	10	0.5	0.3
2	0.01	2	0.5	0.3
3	0.01	1	0.5	0.3
4	0.01	0.1	0.5	0.3
5	0.01	0.01	0.5	0.3
**D_2_ Investigation**				
Test Point	C_i_	D_1_	D_2_	D_3_
1	0.01	1	5	0.3
2	0.01	1	1	0.3
3	0.01	1	0.5	0.3
4	0.01	1	0.1	0.3
5	0.01	1	0.05	0.3
**D_3_ Investigation**				
Test Point	C_i_	D_1_	D_2_	D_3_
1	0.01	1	0.5	1
2	0.01	1	0.5	0.3
3	0.01	1	0.5	0.03

**Table 8 polymers-14-00029-t008:** Results of the fiber length characterization with dynamic image analysis.

Parameters	Region	10th Percentile (µm)	50th Percentile (µm)	90th Percentile (µm)
**First Measurement**				
	Granules 1	157.12	313.71	625.36
Setting 3	Sprue	154.11	293.04	495.81
Setting 3	End	146.59	288.57	509.03
**Second Measurement**				
	Granules 2	153.50	322.32	660.98
Setting 5	Region 1	175.82	334.84	571.74
Setting 5	Region 2	146.83	277.03	473.81

**Table 9 polymers-14-00029-t009:** Mean values of fiber length, diameter, and aspect ratio in granules and regions 1 and 2.

Parameters	Region	L (µm)	d (µm)	L/d (-)
	Granules 2	322.32	11.08	29.09
Setting 5	Region 1	334.84	11.76	28.47
Setting 5	Region 2	277.03	11.22	24.69

## Data Availability

Not applicable.

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
