# Peer review of "Comparison of Real and Simulated Fiber Orientations in Injection Molded Short Glass Fiber Reinforced Polyamide by X-ray Microtomography"

_polymers, 2021, doi:10.3390/polym14010029_

Round 1

Reviewer 1 Report

I have read the authors manuscript and have several comments that they need to address before acceptance of their article:

a) Line 64. Please correct Reference numbers.

b) Several typo errors along the manuscript need to be fixed.

c) Line 82: The authors need to provide preliminary information about the tensor components. I ask them to add an image to show this tensorial representation.

d) Please revise line 92. 

f) Eqs. (1), (2) and (3) need to be explained. Also all the parameters need to be defined.
i, j,k =1,2, 3. Notation needs consistency. For instance, in the text you are using aij but in your equations aij.

g)Line 137: Please provide which method is used to transform differential equation into algebraic ones. Is it finite element method? Please provide information.

h) Line 172: Can the authors add a real sample image?

i) Line 186. Please provide the methodology you followed to optimize machine parameters.
The reader need to understand what would happened if for instance,  
Nozzle temperature  is changed from 280 °C to 250°C and so on.

j) Line 223. Based on your experimental tests, did you check fiber orientation symmetry?

k) Please explain quantitatively and qualitatively why Setting 5 is considered optimal ?

l) Line 240: Must be Figure 2 not Figure 1.

m) Line 228: I do not understand why the authors used the Orthotropic 3 if they are considering isotropic fiber sample distribution. Please explain.

n) Line 292: Please provide information about the error convergence value based on your mesh used.

o) Line 374: Please revised Ref. [42] and [43] to connect your findings with the representation of the material properties.

p) Line 432: The authors need to provide an image that must show the axes representation of the Tij tensorial components.

q) Line 660: The authors need to connect their results with possible 
MRD assumptions. Please do your best to provide insightful information regarding model limitations to described this very complicated nonlinear problem.

r) Figure captions need to be revised. Captions must provide insightful information regarding the image so that the reader do not need to look at information into the manuscript paragraphs.

Author Response

Dear Editor,

thank you very much for your comments. Below is a listing of the responses to your comments:

a) Line 64. Please correct Reference numbers.

- References have been added to the List of References and links have been updated.

b) Several typo errors along the manuscript need to be fixed.

- The text was extensively revised and several errors were corrected.

c) Line 82: The authors need to provide preliminary information about the tensor components. I ask them to add an image to show this tensorial representation.

- A figure (Figure 1) and a reference to it in the text section explaining the tensor representation have been included.

d) Please revise line 92. 

- The paragraph around line 92 was revised.

f) Eqs. (1), (2) and (3) need to be explained. Also all the parameters need to be defined.
i, j,k =1,2, 3. Notation needs consistency. For instance, in the text you are using aij but in your equations aij.

- The formulas and text illustrating and explaining the underlying models have been extensively revised. Some formulas have been replaced for better comprehensibility. Furthermore, all parameters listed in the forms are now defined in the text.

g) Line 137: Please provide which method is used to transform differential equation into algebraic ones. Is it finite element method? Please provide information.

- According to the official European Moldflow support, the finite element method is used to transform the differential equations into algebraic ones. This information has been inserted in the text in Chapter 1 in line 158.

h) Line 172: Can the authors add a real sample image?

- A photo of the specimen has been included in Figure 2.

i) Line 186. Please provide the methodology you followed to optimize machine parameters.
The reader need to understand what would happened if for instance,  Nozzle temperature  is changed from 280 °C to 250°C and so on.

- The procedure for finding and setting the optimum process parameters in injection molding has been added to the text in Chapter 2.3.

The effects of process parameter changes were investigated in the tests for this publication and presented in the results of the real tests and.

j) Line 223. Based on your experimental tests, did you check fiber orientation symmetry?

- The symmetry of the fiber orientation was verified and demonstrated both in the specimens for this publication and in previous studies.

k) Please explain quantitatively and qualitatively why Setting 5 is considered optimal?

- Pleas see comment i) for more infomations

l) Line 240: Must be Figure 2 not Figure 1.

- No, Figure 1 (now figure 2), which shows the two regions, is correct. Figure 2 (now figure 3) shows the locations for the µCT measurement in region 2, with which the structures in this area are to be displayed and compared at higher resolution.

m) Line 278: I do not understand why the authors used the Orthotropic 3 if they are considering isotropic fiber sample distribution. Please explain.

- In our investigations, no isotropic fiber distribution is assumed. A description of the influence of the parameter Ci on the isotropy was inserted in chapter 2.8 in lines 386-390.
However, the setting "Orthotropic 3" in Moldflow is not an (orthotropic) material model but a preset which serves to approximate the fourth degree tensor (closure approximation). In line 143 to 145 these closure approximations were named again.

n) Line 292: Please provide information about the error convergence value based on your mesh used.

- The theory of convergences is now described in Chapter 1 in lines 180-183. Settings for the convergence criteria used in the simulations are listed in chapter 2.6 in table 4.

o) Line 374: Please revised Ref. [42] and [43] to connect your findings with the representation of the material properties.

- The explanation respectively the connection is given in an inserted paragraph at the end of chapter 3.7.

p) Line 432: The authors need to provide an image that must show the axes representation of the Tij tensorial components.

- No new picture was inserted here, but reference was made in the text to the figure inserted for comment c) for the tensor values (Figure 1) and the coordinate system in Figure 2.

q) Line 660: The authors need to connect their results with possible 
MRD assumptions. Please do your best to provide insightful information regarding model limitations to described this very complicated nonlinear problem.

- The explanation respectively the connection is given in an inserted paragraph at the end of chapter 3.7.

r) Figure captions need to be revised. Captions must provide insightful information regarding the image so that the reader do not need to look at information into the manuscript paragraphs.

- All figure and table captions have been extensively revised.

Reviewer 2 Report

In this manuscript, µCT was used to verify the simulation results.

Although the influence of model parameters seems still not clearly and Moldflow ® still cannot give an accurate prediction, this paper provides reference for using Moldflow ®.
We recommend that this paper can be published after revision as indicated.

1) Line 64 Reference [190; 191], Line 95 [203]. These references are not available in this paper.

2) Some writing errors, such as: Line 66, 60-70%]; Line 101, Moldflow®® software; Line 385, images reault from.

3) In Table 1 and Table 2, could you explain how to measure the “flow rate”?  Or how is it controlled by an injection molding machine? 

4) Figure 5 and Figure 6, figures are got from XZ or YZ plane. Please explain it in words.

Author Response

Dear Reviewer,

thank you very much for your comments. Below is a listing of the responses to your comments:

1) Line 64 Reference [190; 191], Line 95 [203]. These references are not available in this paper.

- References have been added to the List of References and links have been updated.

2) Some writing errors, such as: Line 66, 60-70%]; Line 101, Moldflow®® software; Line 385, images result from.

- The text was extensively revised and several errors were corrected.

3) In Table 1 and Table 2, could you explain how to measure the “flow rate”?  Or how is it controlled by an injection molding machine? 

- In our experiments, the injection volume flow (flow rate) was calculated and controlled via the translatory movement of the screw. An explanatory sentence on this has been added in chapter 2.3.

4) Figure 5 and Figure 6, figures are got from XZ or YZ plane. Please explain it in words.

- Figures 5 and 6 (new numbers 6 and 7) show the fiber orientations in region 1 and 2 resulting from the manufacturing of the specimens with different process parameters in injection molding.

Here, both the fiber orientations of the 3 specimens in the XZ plane and in the YZ plane are given for comparison. To be able to relate these planes to the specimen, we refer to the coordinate system in Figure 2.

Round 2

Reviewer 1 Report

The authors have responded to all my comments well; therefore, their article can be accepted for publication in its present form.